

# Review article: Current State of Deep Learning Application to Water-related Disaster Management in Developing Countries

Kola Yusuff Kareem[1], Yeonjeong Seong[1], Shiksha Bastola[1] and Younghun Jung[1,*]

[1]Department of Advanced Science and Technology Convergence, Kyungpook National University, Sangju, 37224, Gyeongbuk, Korea.

*Correspondence to*: Jung Younghun (y.jung@knu.ac.kr)

**Abstract.** Availability of abundant water resources data is a great concern hindering adoption of deep learning techniques (DL) for disaster mitigation in developing countries. However, over the last three decades, a sizeable amount of DL publication in disaster management emanated mostly from developed countries with efficient data management systems. To understand the current state of DL adoption for solving water-related disaster problems in developing countries, an extensive bibliometric review coupled with a theory-based analysis of related research documents is conducted from 1993 – 2022 using Web of Science, Scopus, VOSviewer software and PRISMA model schema. Results revealed a 'slightly' increasing trend of DL-based water disaster publication in developing countries ($tau = 0.35$, $p = 0.00045$, Sen-slope, s = 0.00 at confidence level of 95%), as opposed to the 'significantly' increasing trend globally ($tau = 0.910$, $p = 1.72$ e-12, Sen-slope, s = 2.52 at confidence level of 95%). Also, pluvio-fluvial flooding is found to constitute 78% most disaster prevalence and China is the only 'high human development' developing country with an impressive 51% DL adoption rate, due to China's increasing need for AI-based solutions to persistent multiyear severe water stress, climate change, environmental degradation, recurrent flood, and saltwater intrusion into estuaries. COVID-19 among other factors is identified as a driver of DL adoption. Further analysis indicates that developing countries will experience implementation delay based on their low Human Development Indices (HDI) because model deployment in solving disaster problems in real life scenarios is currently lacking due to high cost. Therefore, data augmentation, transfer learning, intensive research, deployment using cheap web-based servers and APIs are recommended to enhance disaster preparedness. Developing countries can explore these solutions to foster inclusion in global DL-based disaster mitigation approaches.

## 1 Introduction

In the last three decades, there has been a remarkable paradigm shift in the way and manner at which computers are given instructions and how computers are expected to execute such instructions. With the evolution of high performing computers equipped with turbo-charged Random Access Memory (RAM) and powerful Graphics Processing Unit (GPU), computing has never been so fast, accurate and computationally efficient. Artificial Intelligence (AI) and Machine Learning (ML) have given birth to an outstanding technique which builds its functionalities like the functioning of biological neurons found in the brain(Shen, 2018; Xie et al., 2021). According to Reinagel (2000), the Claude Shannon's classic 1948 findings on information theory across a noisy channel depicted the transmission of information using telegraph lines by Morse code. A similar




mechanism is observed in biological neurons. By mimicking the way dendrites transmit impulses through the axon to the brain cells, information can be conveyed using cell states of neurons in models' internal architecture, such that information is stored, processed, and outputted in desired format (Sit et al., 2020). This seemingly new technique is called Deep Learning (DL).

Deep learning models are equipped with features which help to harness temporal dependencies available in reliable hydrological time series data. Generally, DL models are perceived as a black box which learns and interprets complex interactions in data to infer scalable and reasonable results, while reducing modeling stress that comes with conventional modelling approaches (LeCun et al., 2015). DL makes use of Artificial Neural Networks (ANN) to achieve this phenomenon. Several years ago, ANNs were developed to imitate the learning capabilities of human beings and animals. Afterwards, as

computers became faster, smarter and more user-definable, machine learning libraries evolved to achieve the possibility of replicating several days of experimentation in minutes. At first, ANNs employed a trial-and-error approach to solve computational problems by memorizing information from data that has been availed. Over time, they became more flexible and adaptable so much that there are multiple kinds of ANN models currently. Little did the world know that that it would become the epicenter of all computing and predictive tasks. Figure 1 shows the similar architectural and functional composition

of ANNs and biological neurons.

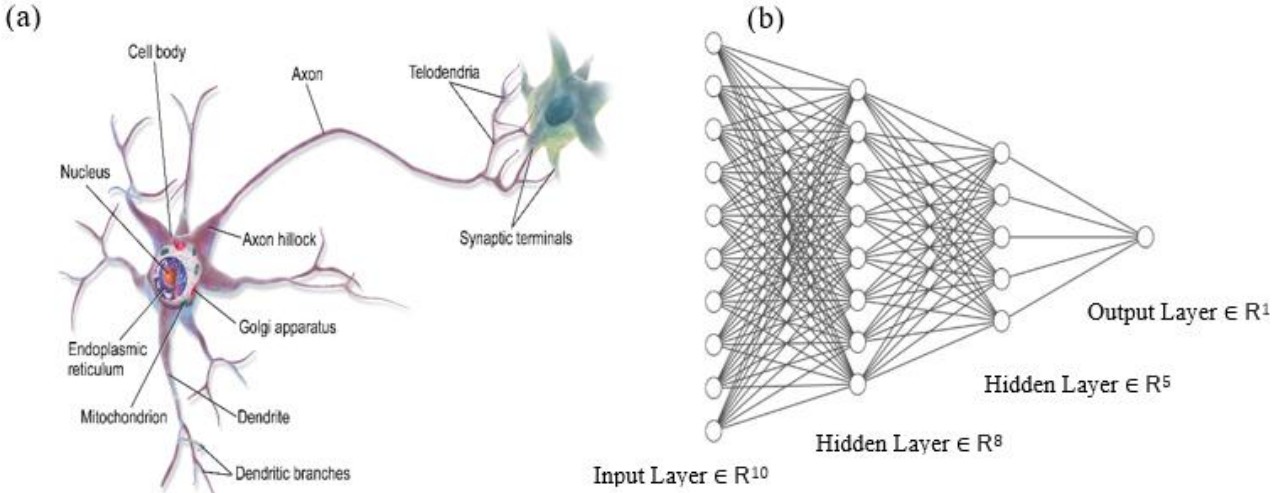

**Figure 1: (a) Multipolar biological neuron; (b) artificial neuron**

Sources: (a) Bruce Blaus Wikimedia Commons (b) authors

Consequently, abundant availability of big data sources, large data storage volumes and excellent sharing features have spurred the adoption of artificial intelligence in many sectors globally. Reinsel *et al.* (2018) projected that the total volume of available global digital data will be 175 zettabytes by 2025. DL models can leverage on the big data repository to proffer lasting solutions to problems that may have appeared insurmountable with use of ordinary physical models. In light of this, applications of DL





have been reported in earth sciences (Reichstein *et al.,* 2019), solar radiation forecast (Ghimire *et al.,* 2019), detection of
arrhythmias in electrocardiograms (ECG) (Oh *et al.,* 2018), stock price prediction (Vidal and Kristjanpoller, 2020), mechanical
tool wear prediction in foundries (Zhao *et al.,* 2017), and several others (Hatami et al., 2018; Kim & Han, 2020; Liu et al.,
2020; Mosavi et al., 2018; Yang et al., 2020) . Hydrological applications of deep learning technique gained momentum in the
last two decades and this has extended to the fields of rainfall-runoff modelling (Kratzert et al., 2018; Ouma et al., 2021);
(Gauch et al., 2021), streamflow and water level prediction for early warning systems (Le et al., 2019; Razavi & Coulibaly,
2013; Shuofeng et al., 2021; Park et al., 2022; Kratzert, et al., 2019, Park et al., 2020; Kareem et al., 2021), water quality
management (Ighalo et al., 2021; Loc et al., 2020), flood susceptibility analysis (Fang et al., 2021; Wang et al., 2020) and
several other interesting fields of hydrology.

Several literature reviews are not conducted on the premise of evidence-based analysis in the selection of research papers for
review (Mosavi et al., 2018; Yang et al., 2020), and this reflects a discrepancy in qualitative and quantitative assessment of
such articles, such that the goal of the literature review might be defeated before writing the manuscript. This raises many
questions about how meticulous or careful the researcher is in defining the selection criteria that culminate into selecting the
resulting papers of study. To solve this problem, we identified DL-based water-related disaster publication that reported hydro-
meteorological datasets from developing countries only, while we ignored author's affiliation because it might be misleading.
A researcher might be affiliated to a research institute in a developed country while he or she uses hydro-meteorological
datasets of a developing country. Therefore, data source was prioritized over affiliation.

Then, we explored research trends in the water-induced disaster in developing countries through a systematic review, identified
research gaps, similitudes, and recommendations for more holistic adoption of DL in these countries. The scope of the study
is streamlined to this region because there is a link between national economic status of every country and interest in adoption
of artificial intelligence, especially DL in water management, climate change studies and water-related disaster risk mitigation
(Pham et al., 2021). The aim of the study is to assess current adoption needs and trends of deep learning technique for water-
related disaster management in developing countries and proffer lasting solutions adapted from developed countries.

## 1.1 Developing countries based on Human Development Index

The Human Development Index (HDI) is computed as the geometric mean of the normalized indices of life expectancy,
education, and income. Contrastingly, the generally acceptable Gross Domestic Product GDP has failed to account for
important good-living metrics such as knowledge base, life expectancy, decent standard of living as explained by the Gross
National Income (GNI) per capita, while considering population density of each country (UNDP, 2020). Consequently, the
HDI measures the growth of a country by considering the freedom and opportunity for people to live the lives they value,
while emphasizing citizen's happiness over raw economic prowess (UNDP, 2020). Although the HDI fails to account for
quality of goods, but it is appropriate for our study because people develop interest in AI when their literacy level is high;
financially capable to purchase computers; and sometimes young. All these factors form the crux of the HDI, making it a
perfect yardstick for selection of study area. Ranks are apportioned to each country based on the HDI values, which range from



0 – 1, relative to other countries. Based on the HDI from the 2020 Human Development Report (HDR) of the United Nations Development Programme (UNDP), countries of the world are categorized into Developed (HDI > 0.8) and Developing (HDI < 0.8) countries, with Norway having an HDI of 0.957; Rank of 1, and Niger with HDI of 0.394; Rank of 189 respectively

(UNDP, 2020). Hence, a developing country is a sovereign nation with a low HDI and a less developed industrial base (Sullivan and Sheffrin, 2003). Table 1 shows a range of global human development indices and components.

**Table 1: Human Development Index**

| Category | Country Count | HDI | HDI Rank | GNI per Capital (2017 PP$) |
|---|---|---|---|---|
| Very high human development | 66 | 0.957 – 0.804 | 1st - 66th | 66,494 – 17,192 |
| High human development* | 53 | 0.796 – 0.703 | 67th – 119th | 26,903 – 13,930 |
| Medium human development* | 37 | 0.697 – 0.554 | 120th – 156th | 4,864 – 3,099 |
| Low human development* | 33 | 0.546 – 0.394 | 157th – 189th | 5,135 – 1,201 |
| Other territories and countries* | 6 | - | - | - | 16,237 – 6,132 |

GNI: gross national income. * Selected developing countries for the study

Source: (UNDP, 2020)

Currently, there are 152 developing countries, which gulp a total population of 6.62 billion, accounting for 85.22% of the world's population (WorldData, n.d.). With abundant water resources dominant in these countries, this is a reasonable population size to assess the early trends of deep learning applications in hydrology for countries with HDI < 0.8. Although, there are controversies about the choice of the word "developed or developing", but other categorizations have similar

meanings. A similar example is the World Bank classification, which considers "upper-middle", "lower-middle", and "low income" or "high human development", "medium human development", and "low human development and others" as developing countries  (UNDP, 2020). All African countries dominate the list of developing countries (Munje & Jita, 2020; UNDP, 2020). At first, through sampling, the authors discovered that DL approaches have not gained significant prominence in Africa, therefore, we extended our scope to developing countries to track the adoption trend, application, and research needs,

knowing fully well that developed countries have set the pace (Razavi, 2021; Shen et al., 2018).





## 2 Methodology

This study considered a bibliometric assessment of publication count, distribution, and growth trend, and identified possible nascent locations that offer promising prospects for future deep learning-focused research. At first, literature search was queried on Web of Science (webofscience.com) and Scopus (scopus.com) databases by specifying abstracts, original research and conference proceedings, and combining germane keywords like 'deep learning', recurrent neural network', 'water-related disaster', 'hydrology', 'streamflow prediction', 'water level prediction', 'disaster', 'flood forecasting', 'flood', 'drought', 'landslide', 'hurricane', 'storm surge' and 'tsunami', with boolean operators of 'OR' and 'AND' and limiting study papers to original published articles in the last three decades (1993 – 2022). Based on the HDI of the country for the study area of each research paper, we identified publication from developing countries and employed the schema of the Preferred Reporting Items for Systematic Reviews and Meta Analyses (PRISMA 2020) model modified by (Page et al., 2021) to guide, streamline and arrive at final selection of forty-nine (49, that is 44 main; 5 in-paper) articles that applied DL models to water-related disaster through duplicate removal, screening, eligibility checks, quantitative and qualitative syntheses.

The PRISMA 2020 model is an improved version of the original PRISMA model developed by Liberati *et al.* (2009) to facilitate synthesis of current state of knowledge, inform future research possibilities, provide in-depth analysis of selected literature materials, and enhance article selection precision by harnessing the benefits of a more meticulous approach of the standard PRISMA 2020 guidelines. The selected articles were obtained based on defined scope of study, relevance to subject matter and were comprehensively studied to arrive at main themes that formed the body of this study. Figure 2 shows the PRISMA 2020 model used for the study and Figure 3 shows the spatial distribution of final selected papers from twelve (12) representative developing countries. Although, the review addressed major thematic areas, but care has been taken to refrain from discussing various neural network architectures as these can be found in numerous publications (Gauch et al., 2021; Kareem et al., 2021; Kratzert et al., 2018b; Razavi, 2021; Shin et al., 2020).

Bibliometric analysis of forty-nine (49) reviewed articles was conducted using Microsoft Excel and VOSviewer version 1.6.17 tools. The latter is a bibliographic assessment tool developed at the University of Leiden, the Netherlands (van Eck & Waltman, 2010). It provides interactive visuals that can be used to map correlations and associations between study features to generate a more accurate representation compared to conventional bibliographic tools like the Multi-Dimensional Scaling (Park et al., 2020).

Finally, we evaluated the study by considering thematic areas of trend analysis, model usage frequency, effect of country's economic development on DL adoption, effect of input data size on model performance, relationship between optimal model and model type, disaster occurrence prevalence, model deployment for solving real life problems in developing countries and conclusion.



**Figure 2: PRISMA model showing selection process of final reviewed articles (data source with country's HDI < 0.8)**



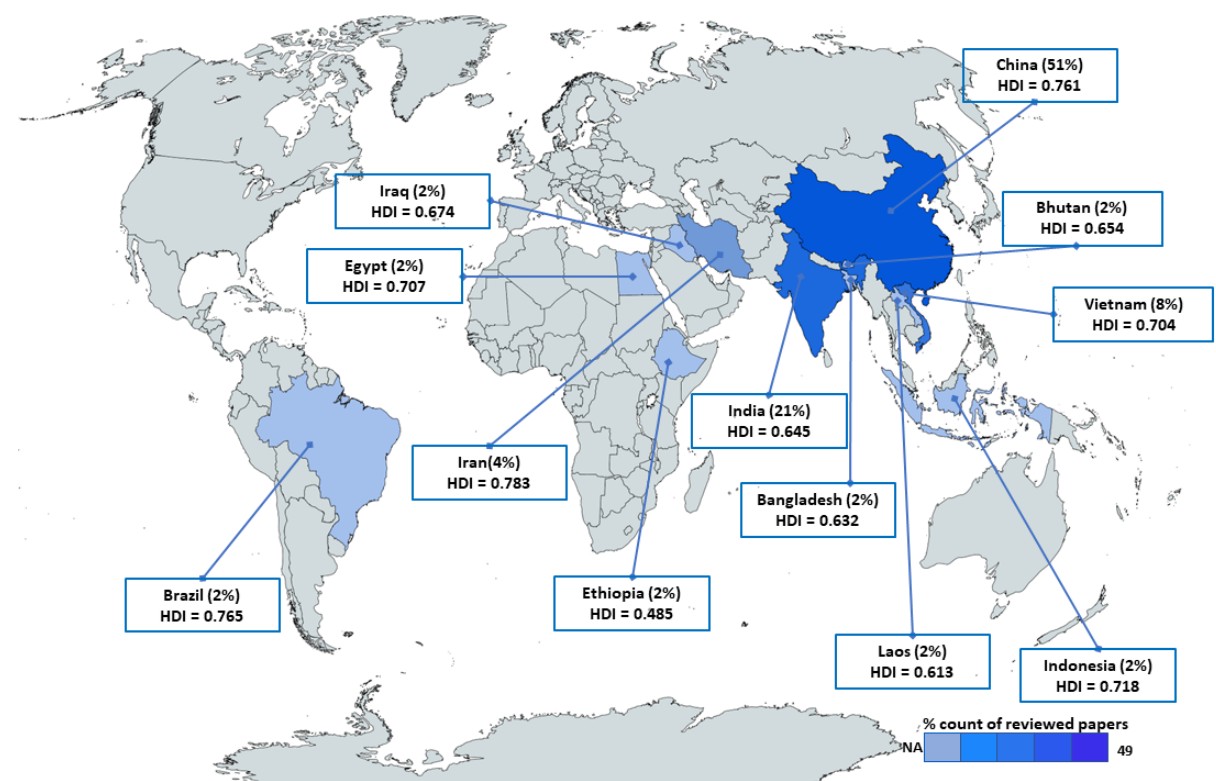

**Figure 3: Spatial distribution of reviewed papers in developing countries**

## 3 Bibliometric Analysis

### 3.1 Trend analysis of water-related disaster in published articles

Publication count analysis of the 49 reviewed articles depicted in Figure 4 identified nine (9) Asian countries, two (2) African countries, and one (1) South American country that published DL-based articles for disaster management in developing countries. It is evident that China recorded highest publication count of 25 (51%); followed by India 10 (21%); Vietnam 4 (8%); Iran 2 (4%); while other 8 countries produced a publication each (2%). In terms of development, DL-focused hydrology articles recorded a spike in countries with high HDI between the range of 0.645 – 0.783, an indication that there is a linear correlation between HDI and computer resources, big data management and willingness to adopt DL for water-related disaster prevention. Therefore, it can be inferred that poor countries will experience a delay in DL implementation for a long period of time.

To understand the global status of DL-focused publication trend in water-related disaster studies between 1993 – 2022 (at the time of writing this manuscript), we compared literature obtained globally with articles from developing countries and analyzed the trend. From Figure 5, it is evident that publication count fluctuated between 2003 and 2017 with intermittent highs and lows. Beyond this period of instability, DL-focused hydrologic studies progressed incrementally to date. Also, statistical trend


analysis of published articles in the world and developing countries in the last 30 years was conducted in python using the
155   Mann-Kendall method (Hussain et al., 2019). The Mann-Kendall method spans between -1 and 1, where values of -1, 0, and
1 signify a perfect declining trend, no trend, and a perfect increasing trend respectively (Newson, 2002). Results showed that
there is a significantly increasing trend in DL-based water disaster publication (*tau* = 0.910, *p* = 1.72 e-12, Sen-slope, s = 2.52)
at a confidence level of 95%. The Sen slope, s value increases at a magnitude of 2.52. This trend can be supported by the huge
global drift towards abundant computer resources, open-access learning platforms and enormous availability of big data
management. Furthermore, it is interesting to attribute the emergence of COVID-19 pandemic to being a driver of AI adoption
and implementation. During the long COVID-19 lockdown exhibited in different countries, people invested ample time to
learn new computer skills while working remotely from their respective homes. By investing adequate time for self-
development programs, water resources engineers and researchers discovered and harnessed amazing benefits of AI, translated
it to research and proffered possible solutions to natural disasters induced by water. This beneficial impact of the pandemic
was supported by Tiamiyu et al. (2021); ); Adelodun et al. (2021), who affirmed that more sensor and satellite technology,
Agriculture 4.0 tool and AI-powered water resources management approaches have been discovered during the first phase of
COVID pandemic due to lockdown and remote operations.

Although a slightly similar trend is observed in developing countries, but with a few variations as illustrated in Figure 6. Mann-
Kendall statistical analysis results depicted a slightly increasing trend in developing countries (*tau* = 0.35, *p* = 0.00045, Sen-
slope, s = 0.00) at a confidence level of 95%. The null Sen slope value is an indication that although there is a slight increase
in trend but the magnitude of change of trend over time is extremely low. Truly. there was a dearth of information in the last
two decades until year 2018, beyond which, DL-focused hydrologic research findings began to gain speed. The progress spiked
in years 2020, 2021 and 2022, with the latter recording more than the previous year's publication just in the first seven months.
The three years witnessed the harrowing effect of lockdown brought about by COVID-19 (Adelodun et al., 2021; Kareem et
al., 2021; Mohan et al., 2021), devastating flood occurrences (Loc et al., 2020; Loganathan & Mahindrakar, 2021), landslide
and other natural disasters (Chen et al., 2021), which required the need for conducting extensive research to mitigate these
disaster occurrences in developing countries.




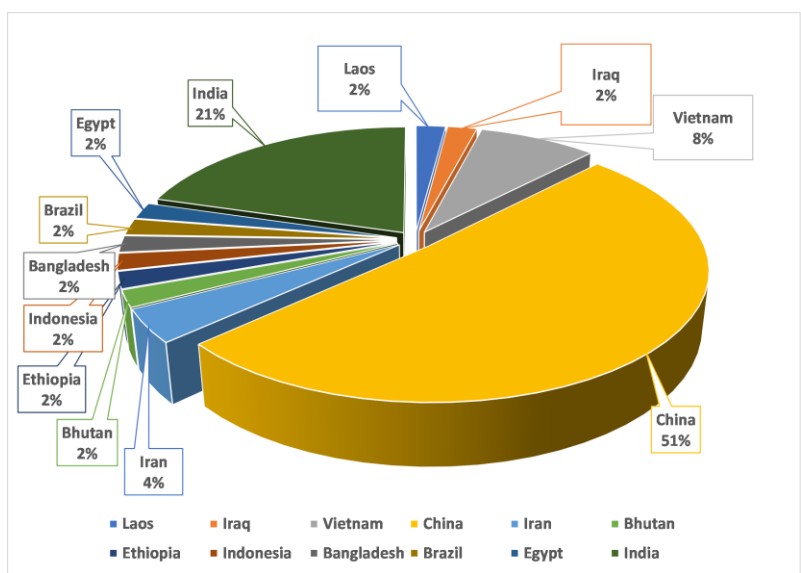

**Figure 4: Publication count by country**

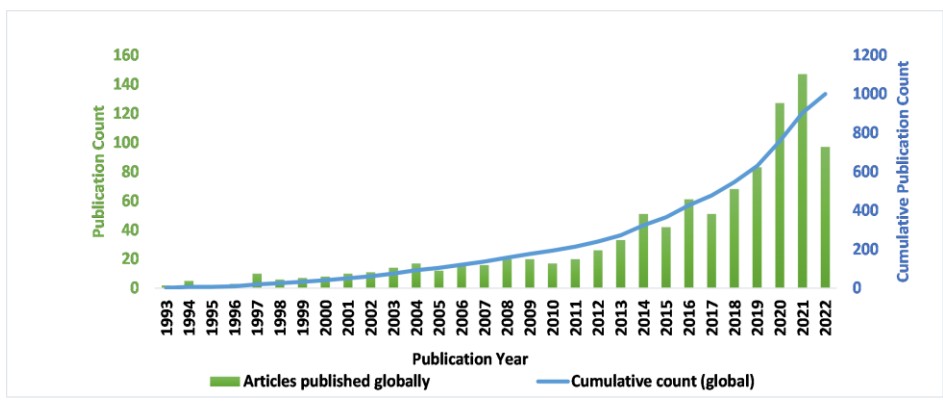


**Figure 5: Global publication trend from 1993 – 2022**

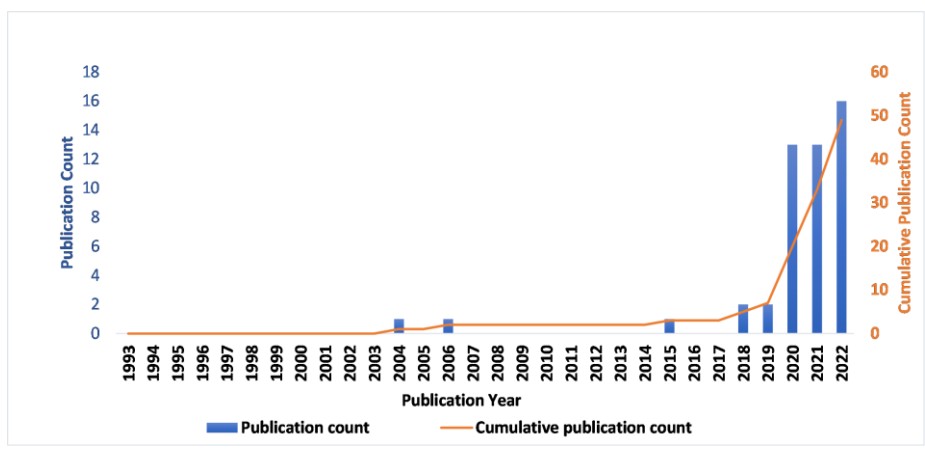

**Figure 6: Publication trend in developing countries from 1993 – 2022**





### 3.2 Co-occurrence analysis of keywords

Co-occurrence analysis of keywords was conducted to identify recurrent keywords in DL studies in developing countries with the use of the VOSviewer tool for frequency and relatedness. Functional words like prepositions, pronouns, conjunctions, and articles like 'the', 'an' etc. were discarded to arrive at 29 items out of 325 that met the minimum 3 occurrences threshold. Figure 7 illustrates the network visualization of keywords. Based on the size of the nodes and total link strength, the top eight

most recurrent keywords are "prediction", ''deep learning", "lstm", "flood forecasting, "machine learning", "model", "precipitation" and "neural network", which formed four clusters of distinct colours. The red cluster (9 items) reveals that deep learning models are essentially used for timeseries analysis, flood forecasting, rainfall prediction in rivers and hydrological systems. The green cluster (8 items) portrayed the misconception that several authors commit by assuming deep learning is similar to machine learning and that long short term memory is a machine learning algorithm, which is not so. The blue cluster

(7 items) reveals that more about predictive applicability of deep learning technique, while the lemon cluster (5 items) shows that models are fitted using ANNs and RNNs. Generally, research articles from developing countries identified deep learning potential in predictive analysis.

### 3.3 Citation analysis

Citation analysis showed that within the last three years, DL findings gained relatively massive prominence and provided scientific knowledge in combating water-related disaster concerns with a total count of 827 citations and an average yearly citation of 221 counts as illustrated in Figure 8. The result identified China as the developing country with the highest citation count (344) and most cited author - (Hu et al., 2018) with 344 citations. Reasons for such a spontaneous drift are due to China's increasing need for AI-based solutions to the persistent multiyear severe water stress and drought (Yu et al., 2014), incessant

climate change and environmental degradation (Henderson, 2004), and recurrent flood and saltwater intrusion into estuaries especially in the southern branch of the Yangtze River (Xue et al., 2009). Co-citation analysis in VOSviewer shows that only 8 articles had cited one another out of the selected 49 articles, with Lee et al. (2019) linking both Hien Than et al. (2021) and Abbas et al. (2020), thereby explaining a very weak research connectivity among authors. Inferentially, studies originating from developed countries have been a good reference material compared to the sparingly few publications emerging from

developing countries. Authors would rather consult literature from developed countries, which had implemented deep learning techniques and currently expanding on ways to improve model accuracy and more efficient model deployment. As this research is the first of its kind, this study can reasonably attribute the cause to possible mistrust that might have arisen from data reliability, procurement, knowledge of subject matter and financial resources capacity predominant in developing countries.





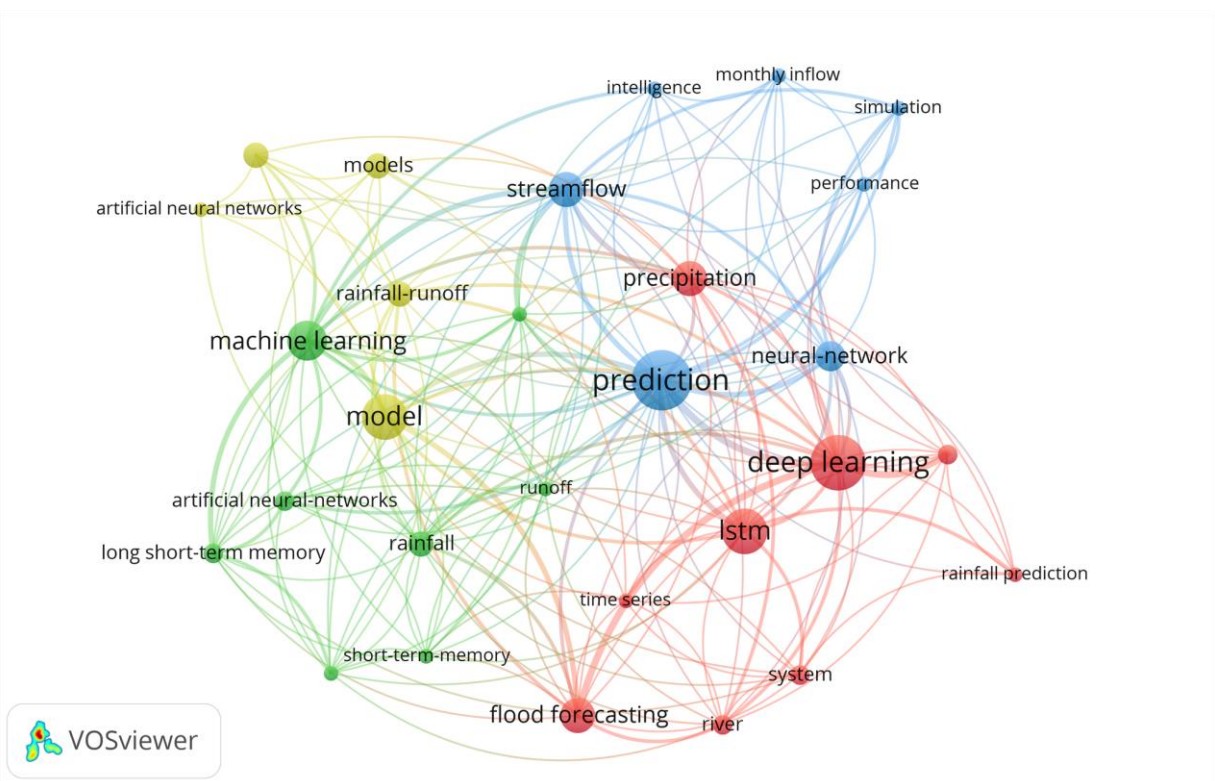

**Figure 7: Network visualization of keywords**

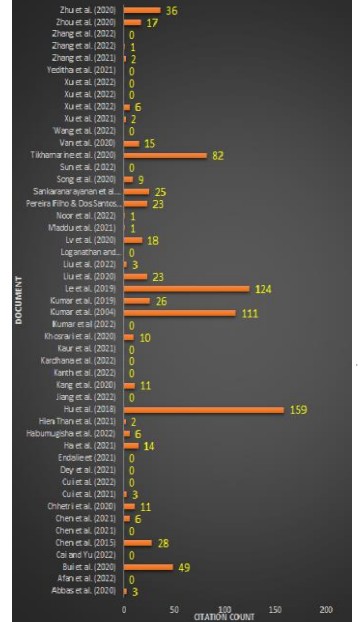

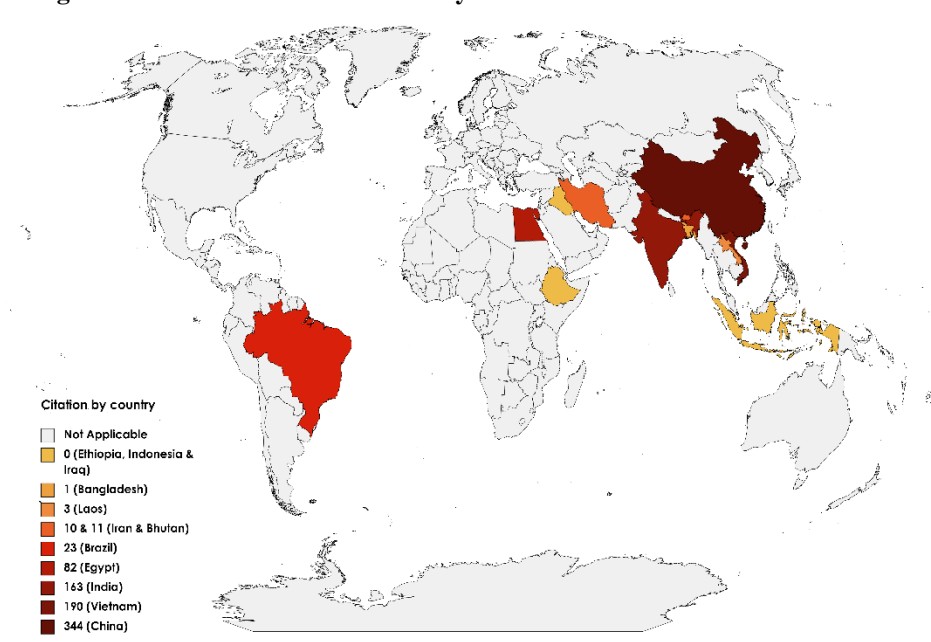


**Figure 8: Citation count of reviewed papers**



## 4 Discussion

### 4.1 Deep learning model usage, economic and geographical significance to adoption

Nine (9) distinct standalone DL models were reported in the reviewed articles with over seventeen (17) hybrid models. Reported hybridization of DL models to improve model performance included integrating swarm intelligence algorithms with
deep neural network (DNN) to classify potential flood risk locations in Muongte district in Vietnam (Bui et al., 2020). (H. Chen et al., 2015) combined DL models with Elman network genetic algorithm to optimize and simulate Hubei Baishuihehe's landslide displacement to yield lowest relative error of 0.44%. According to another research by (C. Chen et al., 2021), developed metaheuristic CNN-imperialist competitive algorithm (CNN-ICA) identified potential snow avalanche events in the west part of Darvan watershed in Kurdistan province. It is apparent that hybrid DL models offer more promising and precise
predictive modelling results by outperforming standalone DL models. (Ha et al., 2021) for integration of El Nino Southern Oscillation with LSTM and (J. Liu et al., 2022) for LSTM-bias corrected hydrometeorological forecast hybrid for flood forecast are also some of the DL hybrid modelling outcomes. Other hybridization approaches explored include the integration of Optimal Variational Mode Decomposition and Improved Hawkins model (OVMD-IHHO-LSTM) to improve LSTM performance for runoff sequence noise reduction (Sun et al., 2022), while (Cui et al., 2021) combined the China's Xinanjiang
physical model with LSTM (XAJ-LSTM) to improve flood forecast accuracy in longer lead times.

The standalone and hybrid LSTM models, being the most used models in this study, were implemented in thirty (30) documents, followed by the CNN and hybrid with thirteen (13) applications, followed by the basic ANN which claimed twelve (12) documents, the GRU and hybrid implemented in seven (7) articles, four (4) MLP articles, three (3) BiLSTM studies, and two (2) articles each for Stacked LSTM, DNN and TCN. Figure 9shows DL model usage in developing countries and the
LSTM model is the most frequently used. Several researchers leverage on the temporal modelling capabilities of the LSTM model to achieve better model performance and propose effective policies to mitigate environmental and disaster risks. Also, the choice of model is highly dependent on modelling needs and approach. Regression tasks constitute about 86% (42 out of 49 articles) of the reviewed articles and it has been reported by several researchers that the LSTM performs best in regression tasks (Razavi, 2021; Kareem et al., 2022; Kareem & Jung, 2021).

China constitutes over 20% of global population and currently boasts of 7% of global water resources reserve. With an HDI of 0.761, China is ranked 87[th] in terms of global development but still reckoned with as a developing country (UNDP, 2020). Based on research contribution from Figure 4, DL adoption has kickstarted in China, thereby setting the pace for other developing countries due to her abundant freshwater in a climate change – induced deluge of precipitation, resulting in devastating flood occurrences and landslides in recent years. Alarming expansion rate of urban population especially in major
cities in China has increased disaster vulnerability, while China's mountainous geo-topographical relief also offered momentum to disaster occurrence. Consequently, the geo-topographical properties and economic level of a country affect DL adoption and implementation as a result of disaster frequency. It is reasonable to state that countries with hilly terrains which aid flash floods require more accurate forecast of impending danger to enhance disaster preparedness.





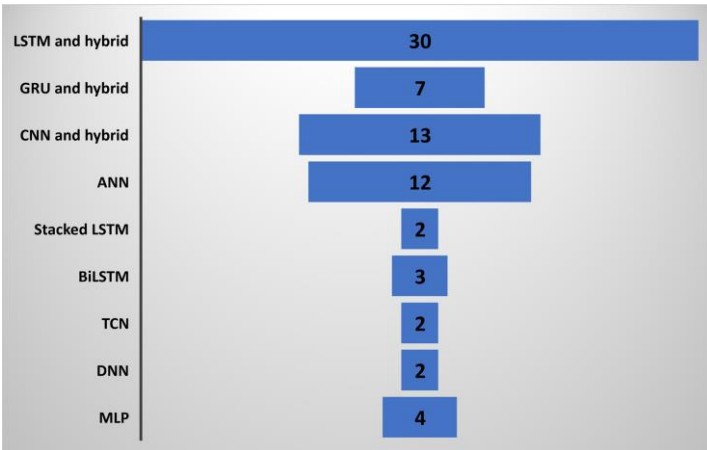

**Figure 9: Model usage**

### 4.2 Effect of data size on model performance

It is quite a herculean task to obtain and compare national water resources data size globally because every country practises diverse data management policies, while some employ decentralized data management systems which thwart unified data collation possibilities for researchers. Therefore, effect of data size on model performance was conducted by quantifying data points for each study, multiplying datasets with number of sampling points or stations to identify optimal performance index for each experiment. In this study, only twenty-one (21) studies which reported Nash Sutcliffe Efficiency (NSE) or Coefficient of Determination ($r^2$) were considered because the two indices are dimensionless entities and will help to ignore numeric errors that might be introduced due to large or small quantities. An example is the error that might be introduced if the root mean squared error (RMSE) values were used because results having larger numeric values would exhibit a higher RMSE, which would translate to low accuracy (Ighalo et al., 2021). An NSE of one (1) depicts a perfect fit, while that of a zero (0) indicates that the model prediction is not any better than the mean of observations, with 0.7 deemed acceptable in hydrology (Razavi, 2021). From Figure 10, it can be observed that there is no clear relationship between performance index and input data size due to the large spatial extent of the study. Although, this shows an anomaly as opposed to the established findings that data size greatly affects performance of DL models (Yetilmezsoy et al., 2011), but it is important to consider several factors that govern performance of neural models like hyperparameter optimization, data wrangling and modeler's domain knowledge. These factors contribute immensely to achieving high predictive model performance, even with limited datasets. Data augmentation techniques and regional modelling which allow for transfer learning can also be applied to generate synthetic data to improve model performance. Also, best performance is not limited to a certain model because the CNN outperformed other models for snow avalanche classification task (Chen et al., 2021), but failed against the hybrid BLSTM-GRU model in rainfall prediction regression task (Chhetri et al., 2020). Therefore, optimal model result is not limited to a certain DL model, but each model performs based on defined modelling objectives.

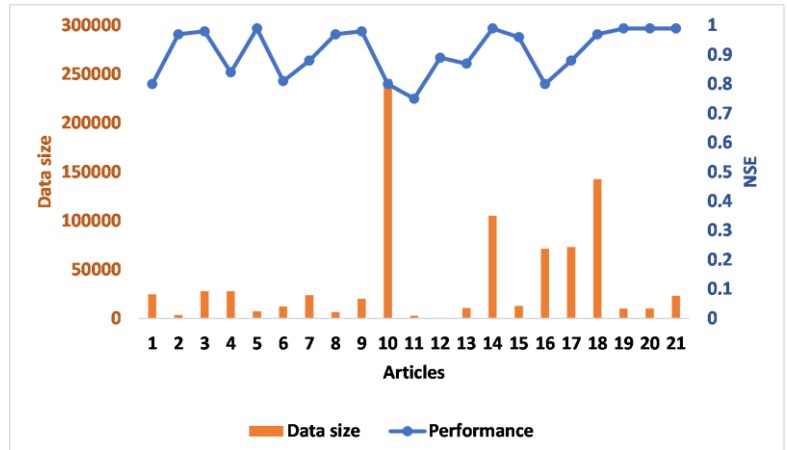

**Figure 10: Effect of data size on model performance**

### 4.3 Deep learning applications for water-related disaster management

Based on this study, five major forms of disaster are recurrent in developing countries and are highlighted as fluvial / pluvial floods, snow avalanche, land subsidence, landslide and drought. Pluvial flooding initiated by extreme rainfall that causes flooding independent of normal river flow, and fluvial flooding events that occur by water level rise in rivers, lakes or streams, thereby overflowing its banks are the most prevalent natural disasters addressed in the articles. Pluvial flooding takes about 78% occurrence compared to others, indicating a dire need for AI-assisted flood risk mitigation approaches in developing

countries. These AI techniques offer modelling potential to inaccessible and dangerous locations with the use of remote sensing and internet of things (IOT). Figure 11 shows the frequency of occurrence of five disaster types identified in reviewed literature. Also, regression tasks have been optimally modelled with LSTM and hybrids while classification tasks achieved best results with CNN and its hybrids. So, choice of model depends mainly on modelling objectives and tasks.

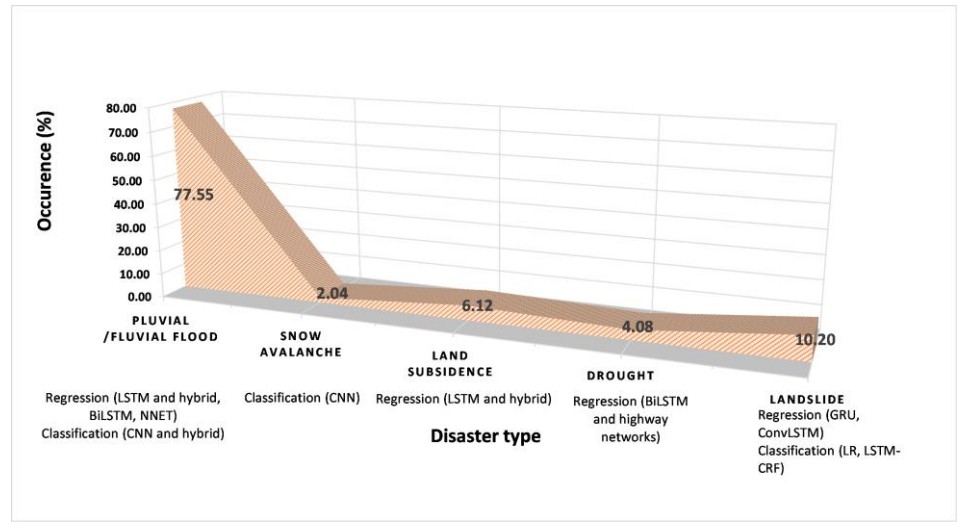

**Figure 11: Major water-related disasters in developing countries**



### 4.3.1 Deep learning application to flood forecasting and rainfall prediction

Flood forecasting and rainfall prediction techniques have improved in the last few years due to the need to address immense economic and environmental losses caused by flood. Thirty-nine (39) out of forty-nine (49) articles explored application of deep learning techniques to flood forecasting, rainfall prediction and adopted various hydrologic modelling approaches like
rainfall prediction (Yeditha et al., 2021; Chhetri et al., 2020; Endalie et al., 2021; Kumar et al., 2019), streamflow forecasting (Abbas et al., 2020; Kumar et al., 2004; Le et al., 2019; Loganathan & Mahindrakar, 2021), flood hazard and severity assessment (Kanth et al., 2022a; Kaur et al., 2021; Khosravi et al., 2020), rainfall-runoff modelling (Van et al., 2020) and flood susceptibility mapping (Bui et al., 2020). Interestingly, this is an indication that developing countries exhibit high flood vulnerability than developed countries, which have embraced better flood protection infrastructure, AI-informed water
dynamics modelling, nature-based ecological solutions, efficient early warning systems, sustainable ecosystem services, sustainable urban design systems, and policies targeted at improving river health and monitoring. A peculiar reason for high flood vulnerability in developing countries is anthropogenic activities such as building housing facilities along floodplains, indiscriminate disposal of solid wastes and wastewater discharge into open waterbodies, industrial effluents disposal, poor land use, illegal farming on plains, poaching of aquaculture, vandalization of floodwater retaining structure and uncultured
cultivation of riparian vegetation. Table 2 shows deep learning application for flood forecasting and rainfall prediction.

**Table 2: Deep learning application to flood forecasting and rainfall prediction**

| Article | Application | Models used | Main findings |
|---|---|---|---|
| 1. Abbas et al. (2020) | Surface and sub-surface flow estimation using environmental time series data and 2D high resolution spatial data | LSTM, HSPF, HRU-based LSTM | Simple LSTM model with one layer performed optimally for surface runoff and flow prediction with lowest MSE = 7.4 x 10-5 m3/s, while HRU-based LSTM model prediction of sub-surface flow recorded optimal results (MSE = 3.2 x 10-4 m3/s) compared to simple LSTM and HSPF flow results. |
| 2. Afan et al. (2022) | Streamflow prediction using linear and stratified sampling techniques | Linear and stratified DL models | Stratified deep learning models improved monthly streamflow prediction accuracy by 7.96 - 94.6 better than linear deep learning models |
| 3. Bui et al. (2020) | Flood susceptibility mapping | Grasshopper, Grey Wolf and Social Spider Optimizations, DNN | Swarm intelligence algorithms improved DNN optimization by |





| | | | identifying potential flood risk locations in Muongte district, Vietnam. |
|---|---|---|---|
| 4. Cai and Yu (2022) | Flood forecasting by hybridization | Hybrid RNN (CNN, LSTM, Bi-LSTM + ARIMAX) | Hybrid RNNs performed optimally among standalone RNNs and the Xinanjiang traditional hydrologic model with optimal NSE = 0.94 and fewest outliers |
| 5. Chen et al. (2021) | Flood prediction | CNN with different batch normalizations | Proposed CNN model showed optimal flood peak and arrival time prediction with 24-hour and 36-hr lead times respectively, for an Internet-of-things-enabled hydrological dataset in the Xixian basin |
| 6. Chhetri et al. (2020) | Rainfall prediction | LR, MLP, CNN, LSTM, GRU and BiLSTM | BiLSTM and GRU layer combination predicted rainfall amount with lowest MSE = 0.93 and R2 = 0.87, which was 41.1% better than the LSTM model. |
| 7. Cui et al. (2021) | Flood forecasting by hybridization for longer lead times | XAJ, LSTM, XAJ-LSTM | Hybrid XAJ-LSTM model effectively improved forecast accuracy in longer lead times. |
| 8. Cui et al. (2022) | Flood forecasting | XAJ, LSTM, LSTM-RED, LSTM-EDE | Proposed exogenous Encoder-Decoder LSTM (LSTM-EDE) overcomes bias problem and predicted flow discharge of Lushui and Jianxi basins optimally than the traditional Xianjiang models and other models |
| 9. Endalie et al. (2021) | Daily rainfall prediction | LSTM, MLP, KNN, SWM, DT | LSTM-based rainfall model achieved best RMSE = 0.01 and is proposed for adoption in rainfall prediction for smart agriculture implementation. |
| 10. Ha et al. (2021) | Streamflow prediction | Stacked LSTM, Cov LSTM encoder-decoder LSTM, | Integration of El Nino-Southern Oscillation (ENSO) data with Hanhou hydrological station data enhanced |



| | | Conv LSTM encoder-decoder GRU | flood forecasting of the Yangtze River basin using deep learning models. |
|---|---|---|---|
| 11. Hu et al. (2018) | Rainfall -Runoff modeling | ANN, LSTM | The LSTM model simulated runoff better than the ANN in both validation and testing datasets. |
| 12. Jiang et al. (2022) | Flood prediction | LSTM, CNN, RF, and MLP | A machine learning (ML) Random Forest outperformed MLPR, CNN and LSTM for farmland flood prediction, reduced computational time, recorded optimal real-time forecasts of water level, and evaluated higher economic loss due to waterlogging for a 100 mm rainffall scenario by coupling AI methods and weather forecast in the Sihu basin. |
| 13. Kang et al. (2020) | Precipitation prediction | ARMA, MARS, BPNN, SVM, GA, LSTM | Height of lowest clouds, pressure tendency, temperature, atmospheric pressure and relative humidity are the most important predictors of precipitation in the Jingdezhen City, China and different number of hidden neurons does not affect LSTM performance. |
| 14. Kanth et al. (2022) | Flood severity assessment using social media streams | ANN, BERT, Bi-LSTM, CNN | Transfer learning using pre-trained models and BERT produced 98% accuracy for predicting flood severity |
| 15. Kardhana et al. (2022) | Water level prediction | ANN, Simple RNN, LSTM-RNN | The LSTM-RNN hybrid maintained $R^2$ prediction accuracy of 0.80 for Katulampa's water level up to 24 h lead time. |
| 16. Kaur et al. (2021) | Early flood prediction using cloud framework | PCA, ANN | ANN predictive algorithm yielded 97.3% sensitivity and future flood |



| | | | stages were forecast using Holt Winter's model |
|---|---|---|---|
| 17. Khosravi et al. (2020) | Spatial flood hazard prediction | CNN | Flood susceptibility mapping using CNN produced an acceptable Area Under Curve (AUC) accuracy of 75% identifying 49% of cities in Iran as highly susceptible to flooding. |
| 18. Kumar et al. (2004) | River flow forecasting | FFN, RNN | Early findings on application of ANNs to monthly streamflow showed that RNNs outperformed FFNs for flow forecasting and so, required further study. |
| 19. Kumar et al. (2019) | Precipitation forecasting | RNN and LSTM | LSTM models validated on different homogeneous rainfall regions in India yielded NSE values between the range of 0.54 – 0.84 regardless of raw data variations. |
| 20. Le et al. (2019) | Flood forecasting | LSTM | Input data type has more effect than data size for better LSTM flood forecasting results when there is a strong linear correlation between input data and target data. |
| 21. Liu et al. (2020) | Streamflow forecasting in different climate zones | RNN, XAJ, LSTM, LSTM-KNN | Analysis of prediction accuracy of models in different climatic zones showed that the KNN algorithm improves the LSTM in streamflow forecasting better than the Xianjiang conceptual model. |
| 22. Liu et al. (2022) | Streamflow and runoff prediction using bias-corrected forecasts | LSTM, Meteo-Hydro-LSTM, ESP-Hydro, Meteo-Hydro | Addition of LSTM model to hydrometeorological forecast (Meteo-Hydro-LSTM) improves forecast skill by a maximum of 25% and average of |



| | | | 6% in the cascade reservoir catchment of Yantan Basin, China |
|---|---|---|---|
| 23. Loganathan and Mahindrakar (2021) | Streamflow simulation and forecasting for early warning systems | NNET and ML models: EXGBDT, DT, KNN, PLS, GLM and PCR | EXGBDT outperformed other six models including the NNET to yield NSE = 0.8, for simulating baseflow, low-flow and high-flow statistics in the Cauvery basin, India |
| 24. Lv et al. (2020) | Discharge forecasting of flood events | LSTMC, BPNNC, LRC | Mutual Information aided LSTM input variable selection and improved its prediction of flow for linear and complex flood systems. |
| 25. Noor et al. (2022) | Water level forecasting | ANN, LSTM, TALSTM ,SALSTM, STALSTM, | Incorporation of attention modules with LSTM improved performance of spatio-temporal attention LSTM (STALSTM) for water level forecasting. |
| 26. Pereira Filho & Dos Santos (2006) | Streamflow prediction | ANN, ARIMA | ANN can be applied to non-linear hydrologic systems because it outperformed ARIMA for streamflow forecasting |
| 27.Sankaranarayanan et al. (2020) | Flood prediction | Deep neural network, KNN, SVM, and Naïve Bayes | ANN performance improved by 40% when either telemetric stage or streamflow was combined with rainfall for flash flood forecasting |
| 28. Song et al. (2020) | Flash flood forecasting | LSTM, LSTM- flash flood (LSTM-FF) | Discharge values enhanced flood prediction by LSTM model for 1 hr lead time while effect reduces with increasing lead time. |
| 29. Sun et al. (2022) | Runoff prediction with optimal variational mode decomposition, improved Harris | BP, LSTM, ELM, PSO-LSTM, HHO-LSTM, IHHO-LSTM, VMD-IHHO-LSTM, OVMD-IHHO-LSTM | Integration of optimal variational mode decomposition (OVMD) and improved Hawkins model improved the LSTM by reducing noise of runoff sequence. |



| | | Hawks algorithm and LSTM hybrid | |
|---|---|---|---|
| 30. Tikhamarine et al. (2020) | Streamflow forecasting | ANN-GWO, ANN, SVR-GWO, SVR, MLR_GWO | The GWO optimized the hyperparameters of SVR and improved prediction accuracy better than traditional SVR, ANN and MLR. |
| 31. Van et al. (2020) | Rainfall-runofff modelling | CNN, LSTM | A 1D CNN model outperformed LSTM, ARIMA, SARIMA and others for regression -based rainfall-runoff modelling of Mekong Delta, thereby indicating the applicability of CNN models. |
| 32. Wang et al. (2022) | Streamflow forecasting with regional characteristics | GRU, RF, SVR | The GRU streamflow forecasting model performed well in almost all seven basins, but poor peak prediction accuracy was recorded as lead time increased. |
| 33. Xu et al. (2021) | Flood forecasting | TCN, TCN (NDVI), LSTM, EIESM, ANN | Temporal Convolutional Network (TCN) with NDVI generalizes and captures rainfall-runoff process more than ordinary LSTM, EIESM and ANN for flood forecast lead times of 1 hr, 6 hrs and 12 hrs |
| 34. Xu et al. (2022) | Rainfall-runoff simulation for short term forecasting | LSTM, PSO-LSTM, ANN, PSO-ANN | Flood forecasting accuracies at different lead times beyond 6 h were improved by using particle swarm optimization - LSTM hybrid. |
| 35. Xu et al. (2022) | Flood prediction | CNN-LSTM, CNN-GRU, SWAT, | Hybridization of CNN with LSTMN or GRU helps to extract local features (CNN) and learn time series dependencies (LSTM and GRU)for predicting monthly discharge in the Xixian basin better than SWAT model |



| 36. Xu et al. (2022) | Monthly streamflow prediction | CNN-GRU, | Deep learning model performance increases with increasing watershed drainage areas. NSE of study areas improved from 0.39 to 0.62 while MRE decreases from 49.9% to 20.9% |
| 37. Yeditha et al. (2021) | Satellite precipitation input for rainfall-runoff modelling | ANN, ELM and LSTM | Optimal LSTM model simulated rainfall-runoff relationships with NSE = 0.87 using two satellite-based precipitation datasets and a ground-based dataset but underestimated peak flood with maximum prediction error of 19.23%. |
| 38. Zhang et al. (2021) | Flood forecasting | MSBP and Random Forest | The Multi-step Back Propagation model predicted flow of the river basin 20 hours ahead with NSE = 0.89 |
| 39. Zhou et al. (2020) | Flood forecasting | NARX, BPNN | Developed NARX model coupled with the unscented kalman filter (UKF)increased reliability of probabilistic flood forecasts and predicted flood better than the BPNN as the forecast horizon increases |

*LSTM: long short term memory, HSPF: hydrological simulated program-FORTRAN, DNN: deep neural network, CNN:*
*convolutional neural network, GWO: grey wolf optimizer, ICA – imperialist competitive algorithm, LR: linear regression,*
*MLP: multi-layer perceptron, PSO: particle swarm optimization, GRU: gated recurrent unit, BiLSTM: bidirectional LSTM,*
*KNN: k-nearest neighbors, SVM: support vector machine, DT: decision tree, BERT: bidirectional encoder representations*
*from transformers, PCA/PCR: principal component analysis/regression, FFN: feed forward network, NNET: neural network,*
*EXGBDT: extreme gradient boosting decision tree, PLS: partial least-squared regression, GLM: generalized linear model,*
*ARIMA: autoregressive integrated moving average, NAR-MA: non-linear autoregressive-moving average neural network,*
*LSTM-MA: moving average long short term memory, ELM: extreme learning machine. TS: Threat Scores, DBN: Deep Belief*
*Network, CDBN: Convolutional DB Network, CRPS: Continuous Ranked Probability Score, XAJ: Xinanjiang model, LSTMC:*
*LSTM cyclic, LSTMC: LSTM cyclic, LRC: Linear regression cyclic, BPNNC: Back propagation neural network cyclic,*
*STALSTM: spatio-temporal attention LSTM, ANN: Artificial Neural Network, TCN: Temporal Convolutional Network,*
*EIESM: Excess Infiltration and Excess Storage Model, MSBP: Multi-Step Back Propagation, RF: random forest, MARS:*
*Multivariate adaptive regression splines, MLP: Multilayer perceptron, LSTM-RED: Recursive Encoder-Decoder LSTM,*





*LSTM-EDE: Exogenous Encoder-Decoder LSTM, FFN: Feed Forward Network, BHLSTM: Bilstm with highway network, CNN-ICA: CNN and imperialist competitive algorithm, SSO: Social Spider Optimizations, NARX: Non-linear auto-regressive with exogenous input neural network, BPNN: Back propagation neural network, MLPR : Multi perceptron regression, Cascade-parallel LSTM-CRF: Cascade-parallel LSTM Conditional Random Field*

### 4.3.2 Deep learning application for landslide management

Landslide occurrence brought about by precipitation, glacial melt and storms is a prevalent disaster in developing countries, especially in China due to high population density and mountainous terrain (Huggel et al., 2012). Five articles reported DL application to landslide management in

Table **3** with use ranging from using attention-based temporal CNN to improve landslide instability margins from a landslide simulation experiment(D. Zhang et al., 2022), to applying GRU for trend and periodic displacement prediction for the Jiuxianping landslide (Zhang et al., 2022).

**Table 3: DL application for landslide management**

| Article | Application | Models used | Main findings |
| --- | --- | --- | --- |
| 1. Chen et al. (2015) | Landslide deformation prediction | RNN with Elman network | Genetic algorithm optimized-RNN models were effective for simulating Hubei Baishuihehe's landslide displacement with lowest relative error of 0.44% |
| 2. Habumugisha et al. (2022) | Landslide susceptibility mapping | CNN, DNN, LSTM and RNN | DL models showed that slope, rainfall, and distance to faults are the most significant factors affecting landslide events in Maoxian County. |





| | | | |
|---|---|---|---|
| 3. Zhang et al. (2022) | Landslide displacement prediction | ANN, GRU, RF, MARS | Trend and periodic displacement results of the Jiuxianping landslide by GRU produced optimal results with fewest outliers. |
| 4. Zhang et al. (2022) | Landslide risk prediction | LSTM, GRU, TCN, ConvLSTM,TCN-Attn-RNN, RNN-Attn-TCN | Landslide instability margins (LIMs) generated from TOPSIS-Entropy method for a landslide simulation platform was improved by attention-based temporal convolutional network and recurrent neural network (Attn-TCN-RNN) with NSE = 0.62 |
| 5. Zhu et al. (2020) | Landslide susceptibility prediction | Cascade-parallel LSTM-CRF, MLP, Logistic regression, decision tree | Landslide susceptibility prediction considering 14 environmental factors improved using the developed cascade-parallel LSTM- conditional random field model |



| | | | more than other models |
|---|---|---|---|

### 4.3.3 Deep learning application for snow avalanche mitigation

Snow avalanche, which results from fast movement of snow, ice, soil, rocks, debris and vegetation along a gradient was addressed by only one article from the reviewed papers and presented in Table 4. As the occurrence of massive snow avalanche is prevalent in mountainous regions, therefore, developing countries with low undulating topography seem to experience insignificant effect of snow avalanche, especially in tropical regions. This explains the small amount of publication recorded

in snow avalanche management studies using DL. Furthermore, as snow and mountain hydrology fields keep evolving, researchers and policy makers are making concerted efforts in understanding the hydrology and translating it to research.

**Table 4: DL application for snow avalanche mitigation**

| Article | Application | Models used | Main findings |
|---|---|---|---|
| 1. Chen et al. (2021) | Snow avalanche identification and mitigation | CNN, CNN-GWO, CNN-ICA | Hybrid deep learning and metaheuristic CNN-ICA model yielded optimal predictive performance and identified potential snow avalanche events in the west part of Darvan watershed, Kurdistan province using generated snow avalanche susceptibility maps. |

### 4.3.4 Deep learning application for land subsidence, drought, and water quality management

Drought, which is a result of over-abstraction of groundwater, soil shrinkage, famine, irregular precipitation, and climate change manifests as land subsidence over time. Sadly, only four (4) of land subsidence DL articles have been reported in developing countries. One of such is the findings of Kumar et al. (2022) which proposed stacked LSTMs and Vanilla LSTMs as a better substitute to conventional land subsidence methods for predicting land deformations at 14 locations at Jharia coal

fields, India. Drought assessment method through ground water level monitoring of the Varuna River in India was studied by (Dey et al., 2021) using annual average of temperature, precipitation, relative humidity, ground water level and actual evapotranspiration to analyze the interrelationship that exists between climate variables and ground water level fluctuations. A summary of DL models' application to land subsidence in developing countries is presented in Table 5.



**Table 5: DL application for land subsidence, drought and water quality management**

| Article | Application | Models used | Main findings |
|---|---|---|---|
| 1. Kumar et al (2022) | Prediction of land subsistence | Vanilla and stacked LSTMs | Stacked LSTM prediction of land subsidence values shows an accuracy of 95% indicating DL model's applicability. |
| 2. Maddu et al. (2021) | Land surface temperature prediction of coastal cities | ANN, LSTM, LSTM-BiLSTM | RNN and hybrid LSTM-BiLSTM forecast surface temperature with final mean NSE = 0.88 across five cities in India, to mitigate risks associated with global warming, heat waves and biodiversity loss. |
| 3. Hien Than et al. (2021) | Water quality classification and performance evaluation | LSTM, ARIMA, NAR-MA, LSTM-MA | Chemometric and DL techniques enhanced forecast of water quality indices of the Dong Nai River using hybrid LSTM-MA, which outperformed ARIMA, NAR and LSTM. |
| 4. Dey et al. (2021) | Groundwater level monitoring | BiLSTM - LSTM ensemble, BiLSTM, BHLSTM | Future groundwater water level depletion in the Varuna River basin, Uttar Pradesh, India projects best |





possible drought conditions using stacked layers of BiLSTM improved with highway network and calls for more water sustainable policies.

### 4.4 Model deployment and Explainability for solving real-life problems

Model deployment is the final stage of every AI task, and it requires that models are deployed to solve real-life problems. It may be performed in diverse environments and integration is always done with the use of an Application Programming

Interface (API). In this study, there is no reported case of final model deployment to solve real life scenario in developing countries. We may attribute this to the current evolutionary stage of deep learning in developing countries. A model can only be operational if it runs on APIs and web-based platforms to generate policies with resounding precision. Model building can be resource-intensive, but deployment helps to generate return of investment. After a successful deployment, routine maintenance must be implemented to eradicate outliers, noises and the model may be retrained on new data for better

generalization. Also, more research must be conducted to explain the internal architecture of models as opposed to the general belief of being a 'black box'.

Finally, DL approaches require a lot of computational resources in terms of high computing systems, excellent GPU capabilities and speed, which are reportedly deficient in developing countries, especially in Africa (Munje & Jita, 2020). It is appalling to know that some countries in Africa still do not have access to affordable computer systems, thereby making

technology and knowledge transfer a mirage. As opposed to common practices in the USA and Republic of Korea, where students are introduced to computing and programming at a tender age to widen their horizon and explore potentials of the next Albert Einstein, such is not the case in developing countries. One then wonders what becomes of children learning ordinary data processing with placards and diagrams painfully drawn on chalkboards in dilapidated walls of schools, with no possibility of ever owning a device in their lifetime. The most significant reason causing the setback is the huge financial

commitment coupled with efficient data management that comes with the new technology. Amongst other reasons are lack of knowledge and technical knowhow, insecurity, poor internet, and data acquisition. Interestingly, this might change in the next few decades because national policies attempting to integrate robotics and artificial intelligence into several sectors of economy of developing countries are currently being considered and implemented in phases.

**5.0 Conclusion**



Thematic areas from literature which used hydrological datasets of developing countries were selected to assess the current state of deep learning adoption for disaster management in developing countries. An extensive theory-based bibliometric analysis addressing publication and citation count, keyword co-occurrence, model usage and eco-geographical significance, input data relationship with model performance, major bottlenecks and model deployment problems affecting the adoption of

DL in developing countries was studied. Statistical trend analysis by the Mann-Kendall method revealed a 'slightly' increasing trend of DL-based water disaster publication in developing countries ($tau = 0.35$, $p = 0.00045$, Sen-slope, s = 0.00 at a confidence level of 95%), as opposed to the 'significantly increasing trend' globally ($tau = 0.910$, $p = 1.72$ e-12, Sen-slope, s = 2.52 at a confidence level of 95%), indicating slow adoption rate in developing countries. For both cases, DL-based disaster research increased steadily in the last two decades due to the global paradigm shift to data-driven analysis, abundant computer

resources, open access learning platforms and big data management. Developing countries experienced a similar trend as DL adoption spiked in 2020 and 2021 because of COVID-19 lockdown effects, devastating flood occurrences, landslide, and other natural disasters, which required the need for conducting extensive research to mitigate risks and losses. Also, it was discovered that five major natural disasters – pluvio-fluvial flooding, snow avalanche, land subsidence, drought and landslide are prevalent in developing countries, while pluvio-fluvial being about 78% most prevalent. Recurrent flash floods and landslides caused by

irregular rainfall pattern, abundant freshwater and mountainous terrains attributed China (out of 12 developing countries as the only high human development developing country with an impressive DL adoption rate of 51% publication count.

Further analysis indicates that economically disadvantaged countries will experience a delay in DL implementation based on their HDI because DL implementation is capital-intensive. COVID-19 among other factors is identified as a driver of DL adoption. Although, the Long Short Term Model (LSTM) model is the most frequently used, but optimal model performance

is not limited to a certain model. Each DL model performs based on defined modelling objectives. It was discovered that final model deployment in solving disaster problems in real life scenarios is currently lacking in developing countries. We hereby recommend data augmentation and transfer learning to solve data management problems in ungauged watershed prevalent in developing countries as implemented by Rasheed et al. (2022), Kratzert et al. (2019), and Kratzert et al. (2018) using the CAMELS dataset in the contiguous USA. Intensive research, training, innovation, deployment using cheap web-based servers,

APIs and nature-based solutions are encouraged to facilitate speedy adoption of DL and enhance disaster preparedness in developing countries. We are optimistic that the findings of this study will provide adequate information, initiate speedy DL adoption, become a veritable reference material, and provoke stellar research thoughts towards DL implementation in developing countries.

**Authors' Responsibilities**

Kola Yusuff Kareem: Conceptualization, Methodology, Analysis, Writing – original draft, Review and Editing. Yeonjeong Seong: Methodology, Writing, Analysis. Shiksha Bastola: Methodology, Writing, Analysis. Younghun Jung: Methodology, Review and Editing, Funding Acquisition, Resources Investigation.

**Competing Interest**



The authors declare that they have no conflict of interest.

**Acknowledgement**

This research was supported by a grant (2022-MOIS61-001) of Development Risk Prediction Technology of Storm and Flood for Climate Change based on Artificial Intelligence funded by Ministry of Interior and Safety (MOIS, Korea).

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
