# Peer review of "Review article: Current State of Deep Learning Application to Water-related Disaster Management in Developing Countries"

_Natural Hazards and Earth System Sciences, 2022_

## Author Comment (AC1)

**AC:**      **Authors' Response to Reviewer**

Respected Anonymous Reviewer,

The authors of this manuscript appreciate your invaluable suggestions and comments for further improvement of our manuscript. All comments are very useful. We have read all comments carefully and revised our manuscript according to your suggestions (changes are presented in red color in the manuscript). Detailed responses to the comments are presented below.

1. Most of the Introduction describes the development of deep learning methods, lacking the necessity of developing the review of water-related disasters.

   The Introduction section has been re-written to reveal the need for the research. The last paragraph has been structured to indicate the need, scope and aim of the study – Line 91-94.

   In summary, the authors discovered that developed countries have been researching and adopting more data-driven approaches (specifically deep learning techniques) in mitigating disaster risk through computer vision, flood segmentation tasks, prediction, clustering etc. Only few research papers and reports emanated from developing countries. This created a research need to study the reasons for such a slow adoption, trends, and drivers of deep learning (DL). Therefore, the first paragraph in the Introduction section addressed evolution of AI; second paragraph focused on recent big data availability, while the last paragraph presented the research approach. We briefly discussed the UNDP's Human Development Index to guide our choice of countries ranked as developing countries. Thank you.

2. The selection process of final reviewed articles (Figure 2) is not clear. I am confused about the "records excluded (n=16)" and "Abstract relevance".

   Thank you for raising this comment. After we had submitted the article, we discovered that we did not include the reason for excluding the sixteen (16) articles in our PRISMA schema. The criterion for removal was data source. After the first filtering, we checked study area reported in all ninety-eight (98) articles and removed 16 papers that used hydrological datasets from developed countries (countries with Human Development Index > 0.80 according to UNDP), because we focused on articles from developing countries. We have updated the reason for exclusion in the new draft – Figure 2.

   Abstract Relevance: Then, the authors retrieved and studied eighty-two (82) articles fully. We identified and excluded thirty-eight (38) articles that do not address water-induced disaster but managed to form part of our query output from search databases. We have edited the "Abstract Relevance" reason for clarity in the new manuscript draft – Figure 2. Thank you for this wonderful observation.

3. I suggest to provide the final search keywords with "OR" and "AND". For example, ("flood" OR "drought") AND ("deep learning").

   Thank you for this excellent suggestion. We have included the query from Web of Science as requested - **Line 118 - 110**
   *(TS=("Deep learning" AND "developing countries" OR "recurrent neural network" OR "water-related disaster" OR "hydrology" OR "streamflow prediction" OR "water level prediction" OR "disaster" OR "flood forecasting " OR "flood" OR "drought" OR "landslide" OR "hurricane", "storm surge" OR "tsunami"))*

4. Figure 9 is not clear. – **Figure 9**

Figure 9 shows the chart of models used in our reviewed articles. We have edited the figure caption and the axis title for more clarity.

We identified and counted different deep learning models (not machine learning algorithms) that were used in the reviewed articles to portray how the number of times these models had been used. For example, LSTM and hybrids include LSTM with other DL model ensembles, were applied in thirty-two (32) articles of the reviewed papers. This output is only limited to the scope of our work and our methodology. This was sufficient to track country location and to subsequently categorize the country into developing and developed. Thank you.

5. After the inspection of Table 2-5, all the studies are carried out in recent 4 years, excluding Kumar et al. (2004), Pereira Filho & Dos Santos (2006), and Chen et al. (2015).

Thank you for your wonderful observation. Although deep learning research was first implemented and published by Alexey and Lapa in 1967 (Ivakhnenko et al., 1967), it evolved and gained momentum in water resources and environmental studies only in recent years as a subfield of machine learning. The early implementation was done by developed countries until lately when developing countries decided to adopt the technique. Developed countries had been benefiting immensely from this paradigm shift. For the scope of our work, the HDI criterion helped to track specific research outputs from developing countries and publication dates were reported as stated. Evidently, a larger percentage of all publications in the academic space in the last three decades had been published in developed countries. Our approach might not be exhaustive and there might be very few articles that might miss the designed query but summarily, we believe that this research presents a vivid representation of the trend and an important point of reference for future research.

6. Section 4.2: the effect of data size in different disaster maybe different. For example, we cannot provide a larger number of samples in landslide field.

The authors of this manuscript clearly agree with the reviewer that effect of data size of different disasters may be different. Similarly, the authors had established the same fact that due to the large spatial scope of the study (developing countries) and uncertainties due to heterogeneity, the effect of data size on model performance does not show a clear relationship. Based on the results of the study, it is hard to validate the general findings of the Power Law: "more data, better performance" as established by (Alom et al., 2019; van Essen et al., 2015; Yetilmezsoy et al., 2011) . This raises the question of how much data is enough to obtain optimal modeling results? Although the concept of autocorrelation can select appropriate input variables for better prediction, but the DL field is still evolving and more interesting areas are yet to be explored. In our manuscript, we identified hyperparameter tuning and efficient data cleaning as better factors for improving model performance.
We have edited the previous manuscript for more clarity – Section 4.2 Line 274 - 279

7. Section 4.3.2: In my opinion, if we include the landslide hazard in the review, there are lost of papers that using deep leaning methods in this field.

If the query search is extended to water-related disaster globally or in developed countries, landslide studies and other pertinent disaster studies can be captured. We showed this when we compared publication trend in developing countries and global trend  – Figures 5 and 6. Thank you.

8. This manuscript misses many related papers, such as:

- Wang, Y., Fang, Z., Hong, H., Peng, L., 2020. Flood susceptibility mapping using convolutional neural network frameworks. Journal of Hydrology, 582, 124482

- Fang, Z., Wang, Y., Peng, L., Hong, H., 2021. Predicting flood susceptibility using LSTM neural networks. Journal of Hydrology, 594, 125734.

- Xu, S., Niu, R., 2018. Displacement prediction of Baijiabao landslide based on empirical mode decomposition and long short-term memory neural network in Three Gorges area, China. Computers & Geosciences, 111, 87-96.

- Van Dao, Dong, et al. "A spatially explicit deep learning neural network model for the prediction of landslide susceptibility." Catena 188 (2020): 104451.

- Shahabi, Himan, et al. "Flash flood susceptibility mapping using a novel deep learning model based on deep belief network, back propagation and genetic algorithm." Geoscience Frontiers 12.3 (2021): 101100.

Although the first two papers were studied during the literature review part of the work to provide some background knowledge for the study – Line 61, we agree that they should form part of the main reviewed articles because the studies utilized DL models, data sources were from developing countries, and they fall within our research scope. Therefore, we have extensively reviewed all recommended articles based on our methodology. We have effected changes in the updated manuscript. Thank you.

9. I cannot find any review of flood mapping in this manuscript.

We reviewed flood mapping articles and summarized in Table 2 (Afan et al., 2022; Kanth et al., 2022; Khosravi et al., 2020; Xu et al., 2021). Based on your recommendation, we added more flood mapping articles that met the scope of our study (Fang et al., 2021; Shahabi et al., 2021; Wang et al., 2020).

Thank you for the excellent addition.

10. Section 4.3: it would be better to provide some comparison and summary work for these studies, not only describe each paper standalone.

Thank you for your constructive comments. We analyzed the reviewed papers by discussing the main applications and DL models that the authors reported in their study. We summarized and grouped the applications into: (i) flood mapping, forecasting and rainfall prediction; (ii) landslide susceptibility mapping; (iii) snow avalanche prevention; (iv) land subsidence and drought.

**New edits**

6.74 billion..... and 85.43% (new update from data source) – Line 81

trends  in (word repetition) - Line 94

Hussain and Mahmud (citation edited) - Line 154

Removed Figure 4: Publication count by country (repetition of Figure 3) - Line 176

Removed "Water management" (not necessary) – Section 4.3.4. Thank you.

**References**

Afan, H. A., Yafouz, A., Birima, A. H., Ahmed, A. N., Kisi, O., Chaplot, B., & El-Shafie, A. (2022). Linear and stratified sampling-based deep learning models for improving the river streamflow forecasting to mitigate flooding disaster. *Natural Hazards*, *112*(2), 1527–1545. https://doi.org/10.1007/s11069-022-05237-7

Alom, M. Z., Taha, T. M., Yakopcic, C., Westberg, S., Sidike, P., Nasrin, M. S., Hasan, M., van Essen, B. C., Awwal, A. A. S., & Asari, V. K. (2019). A state-of-the-art survey on deep learning theory and architectures. In *Electronics (Switzerland)* (Vol. 8, Issue 3). MDPI AG. https://doi.org/10.3390/electronics8030292

Fang, Z., Wang, Y., Peng, L., & Hong, H. (2021). Predicting flood susceptibility using LSTM neural networks. *Journal of Hydrology*, *594*. https://doi.org/10.1016/j.jhydrol.2020.125734

Ivakhnenko, A. G., Lapa, V. G., & Valentin Grigor′evich Lapa. (1967). Cybernetics_and_Forecasting_Techniques. *American Elsevier Publishing Company*, 168. https://books.google.co.kr/books?id=rGFgAAAAMAAJ

Kanth, A. K., Chitra, P., & Sowmya, G. G. (2022). Deep learning-based assessment of flood severity using social media streams. *Stochastic Environmental Research and Risk Assessment*, *36*(2), 473–493. https://doi.org/10.1007/s00477-021-02161-3

Khosravi, K., Panahi, M., Golkarian, A., Keesstra, S. D., Saco, P. M., Bui, D. T., & Lee, S. (2020). Convolutional neural network approach for spatial prediction of flood hazard at national scale of Iran. *JOURNAL OF HYDROLOGY*, *591*. https://doi.org/10.1016/j.jhydrol.2020.

Shahabi, H., Shirzadi, A., Ronoud, S., Asadi, S., Pham, B. T., Mansouripour, F., Geertsema, M., Clague, J. J., & Bui, D. T. (2021). Flash flood susceptibility mapping using a novel deep learning model based on deep belief network, back propagation and genetic algorithm. *Geoscience Frontiers*, *12*(3). https://doi.org/10.1016/j.gsf.2020.10.007

van Essen, B., Kim, H., Pearce, R., Boakye, K., & Chen, B. (2015, November 15). LBANN: Livermore big artificial neural network HPC toolkit. *Proceedings of MLHPC 2015: Machine Learning in High-Performance Computing Environments - Held in Conjunction with SC 2015: The International Conference for High Performance Computing, Networking, Storage and Analysis*. https://doi.org/10.1145/2834892.2834897

Wang, Y., Fang, Z., Hong, H., & Peng, L. (2020). Flood susceptibility mapping using convolutional neural network frameworks. *Journal of Hydrology*, *582*. https://doi.org/10.1016/j.jhydrol.2019.124482

Xu, Y., Hu, C., Wu, Q., Li, Z., Jian, S., & Chen, Y. (2021). Application of temporal convolutional network for flood forecasting. *Hydrology Research*, *52*(6), 1455–1468. https://doi.org/10.2166/NH.2021.021

Yetilmezsoy, K., Ozkaya, B., & Cakmakci, M. (2011). Artificial intelligence-based prediction models. *Neural Network World*, 21(3), 193–218. https://hero.epa.gov/hero/index.cfm/reference/details/reference_id/975279

---

## Author Comment (AC2)

AC:     Authors' Response to Reviewer II

Respected Anonymous Reviewer,

The authors of this manuscript appreciate your invaluable suggestions and comments for further improvement of our manuscript. All comments are very useful. We have read all comments carefully and revised our manuscript according to your suggestions (changes are presented in green color here and in the manuscript). Detailed responses to the comments are presented below.

> While the manuscript exhibits strong writing skills, I feel it is necessary to recommend its rejection at this stage based on the following key points:

> 1.  The authors have attempted to explore the use of deep learning for water-related disaster management in developing countries. However, the manuscript falls short in substantiating the need for such a review. It is essential that the authors synthesize existing review studies to identify potential gaps in the current body of knowledge, which could underscore the need for this new review. It is concerning that the cited work of Sit et al. (2020) already provides a detailed exposition of the state-of-the-art, and the authors' approach seems to mirror the structure of this previous study. Therefore, the unique contribution of the current review remains unclear

The Introduction section has been rewritten to reveal the need for the study and has been elaborately discussed in the new draft. The new edit can be found in the last paragraph that explains the need, scope and aim of the study – **Line 116-119.** Here is a copy:

*"Over the last three decades, data-driven techniques aimed at mitigating water disaster risk have gained prominence in developed countries but sadly, few publications emanated from developing countries. To understand the current state of DL adoption for solving water-related disaster problems in developing nations, we identified DL-based water disaster publication that reported hydro-meteorological datasets from developing countries."*

In summary, the authors discovered that developed countries have been researching and adopting more data-driven approaches (specifically deep learning techniques) in mitigating disaster risk through computer vision, flood segmentation tasks, prediction, clustering etc. Only few research papers and reports emanated from developing countries. This created a research need to study the reasons for such a slow adoption, trends, and drivers of deep learning (DL). Therefore, the first paragraph in the Introduction section addressed evolution of AI; second paragraph focused on recent big data availability, while the last paragraph presented the research approach. We briefly discussed the UNDP's Human Development Index to guide our choice of countries ranked as developing countries. Thank you.

The unique contribution of this research is to provide scientific background for the slow adoption of deep learning techniques for water-disaster management. Also, the findings of this research aimed to provide a reference guide to water professionals focused on developing countries through elaborate bibliometric and trend analysis, and tabulated results of reviewed articles. The methodology and results of the study are different from Si et al. (2020) who focused on discussing internal architectures of the AI models and how to utilize such AI methods for future water resources challenges. Our approach is more of a socio-hydrological approach which considers the interrelationship between Human Development Indices (HDI) of each country, availability of computer resources, and level of education. Our study advocates the need

for DL adoption in developing countries and opines opportunities for data-scarce regions (typical of developing countries) to foster inclusion in AI-powered disaster management practices globally.

2. The decision to concentrate the review on developing countries could be deemed justified if the authors were able to identify research gaps that are particularly pertinent to these regions. Simply narrowing down the geographical focus without highlighting specific, unique issues in these areas does not add value or increase the relevance of the review.

Thank you for raising this comment. We have included the justification for selecting developing countries as our scope of study in the new draft – **Line 106-115.**

*"The adoption of modern DL models for water disaster is greatly hampered by lack of structured data ecosystem, skillset, governmental policies, uncertainty, and ethical reasons prevalent in several developing and African regions* (Rutenberg et al., 2021). *This does not aim to discredit the few African countries that are making concerted efforts towards DL adoption* (Pedro et al., 2019; Pillay, 2020; Schoeman et al., 2021). *Common fears being allayed in developing countries is the propensity of robots to replace the unskilled labour force, thereby shifting more investments to advanced economies, creating diverging income levels and widening socio-economic gaps among countries. Cloud-based computing requiring sufficient cyber infrastructure and connectivity is a major cause of concern in developing countries so much that a good percentage of residents do not have access to wireless networks coupled with overbearing cost of procurement* (Mtega et al., 2012; Pillay, 2020). *All these factors account for the demotivation of researchers towards adoption of DL in developing countries and forms the basis of this study"*

3. While the authors broached the topic of using deep learning for water-related disaster management, they did not delve into the concept of disaster management itself. Instead, their focus remained largely on water-related hazards. It is critical for a review of this nature to thoroughly analyze and clarify the key terminologies being used, such as "disaster", "hazards", and "management". This kind of academic rigor enhances the comprehension and utility of the review.

Thank you for this excellent suggestion. The authors of this manuscript have included and clarified the basic terms related to disaster management and included an illustration to identify several disaster classes, events and harm. We have updated these new edits in our draft too– **Line 63-77 and Figure 2.**

*Natural disasters have immensely plagued the Earth and affected the limited water resources resulting in poor environmental quality, poor water quality, pollution, and fatality on several occasions. Disasters ranging from wildfires, floods, earthquakes, storms, volcanic eruptions are triggered by natural hazards, which are defined as events or physical conditions that exhibit potential to injury, fatality, infrastructural damage, agricultural loss, property damage, environmental damage, any kind of harm, and disruption of business activities* (Cutter, 2001; UNISDR, 2009). *Natural hazards can also occur as any form of hazard related to weather patterns and /or physical characteristics of an area* (FEMA, 1997). *Exposure to natural hazards defines the inventory of items like property, people, and systems within the area which hazard events might occur* (Chaudhary & Piracha, 2021). *Exposure to hazards varies with resilience and vulnerability of assets and humans. Vulnerability measures the degree of susceptibility to economic losses, casualties, property losses, and physical injury to assets and humans during exposure to hazards. All these components culminate into Risk, which defines the product of the likelihood of the occurrence of an event and the consequences if the event occurs, as explained in Equation 1. Also, Figure 1 illustrates several*

*classes of disasters and their related main events and harm that exist globally, as modified from* UNISDR (2023).

$$Risk, R = Hazard, H \times Vulnerability, V \qquad (1)$$

[Figure]

Figure 2: Disaster Classifications

4. The review, in its present state, comes across as generic and lacking in specifics. The manuscript offers little insight into research gaps, which raises doubts about the value it could provide to readers. It is vital that the authors convey specific, concrete details and pinpoint gaps in the research to make the review beneficial to its intended audience.

I encourage the authors to consider these points to increase the value of their research, thereby enriching their research contribution and making it more beneficial and relevant to their readers.

Thank you for the comment. The authors of the manuscript have included an extra section 5.0 after the Discussion section to clearly synthesize the research gap and prospects of application of DL to water-induced disaster prevention in future works. **– Line 406-412.**

**Research Gap and Future Prospects**

*Training data for DL techniques for flood susceptibility mapping and flood extent prediction are always improved by the application of flood topographic and hydrological conditioning factors such as digital elevation models (DEM), aspect, slope, total wetness index (TWI), river proximity, soil data, lithology, factor curvature, and stream power index (SPI) etc., to provide more explanatory training data to increase model robustness and learning action* (Fang et al., 2021; Shahabi et al., 2021; Wang et al., 2020). *While some other studies attempted to use different DL models to study landslide displacement, landslide susceptibility mapping, and deformation characteristics such as slip-bending failure and bulging, reviewed articles are limited to only concluding a certain model result performs better. Real-life implementation of such results is deficient, as opposed to applications found in developed nations. This gap in combination with the poor adoption of DL-based water disaster techniques form the basis for reviewing existing literature and proffering possible future prospects to AI / DL researchers and practitioners with a bias for water-related disaster management.*

*The future of DL or AI in general is proposed to be governed by data privacy regulations to set a global standard for AI practices and avert potential AI-induced hazards as captured in the April 2021 European Commission's EU AI Act. This regulatory framework proposes the analysis and classification of different AI system applications according to the corresponding risks posed to providers and users. Such risk mitigation systems aimed to cover associated risks with social scoring, cognitive behavioral manipulations, transparency demands such as disclosure of data source as AI, copyright inclusion, illegal content control and ownership liabilities, with the most recent development being an agreement to host a negotiation meeting for the consideration of the EU AI Act by the Members of the European Parliament in June 2023* (EPRS, 2023).

*When similar regulatory frameworks as this are localized and adapted to developing countries, this will help to build more trust and reliability in adopting DL-based approaches towards water disaster management, thereby providing capacity for innovation.*

*As a double-edged sword, the environmental and computational cost of training very deep neural networks may emit about 650 tons of carbon emission typical of some local flight energy demands. Such emissions may be compensated for by developed countries that embrace green energy and nature-based solutions while developing countries bear the brunt owing to limited investment resources in climate change mitigation actions. On the other hand, through a consensus-based elicitation approach, it was discovered that over 134 targets across all Sustainable Development Goals can be achieved using AI-based (mostly deep learning) approaches, while the accomplishment of about 59 targets may be impeded* (Vinuesa et al., 2020). *This underscores the importance and need for future works to explore DL usage and institute policies that reduce the DL pitfalls in developing countries.*

*Future implementation of DL-based techniques for water-induced disaster must focus on emergency response systems that address pre- and post-damage assessments, spatio-temporal suitability analysis to reveal optimal locations for distribution of relief materials, evacuation of people and properties, monitoring of potential epidemic outbreak, human resources management and business guides, hotspot identification, medical healthcare and prompt drug delivery system to flood victims and other related hazards, while seeing AI as a tool for economic improvement and not a weapon of destruction.*

**References**

Chaudhary, M. T., & Piracha, A. (2021). Natural Disasters—Origins, Impacts, Management. *Encyclopedia*, *1*(4), 1101–1131. https://doi.org/10.3390/encyclopedia1040084

Cutter, S. L. (2001). The changing nature of risks and hazards. In *American hazardscapes: The regionalization of hazards and disasters* (pp. 1–12). Washington, DC: Joseph Henry Press.

EPRS. (2023). *Artificial intelligence act. BRIEFING EU Legislation in Progress*. https://doi.org/PE 698.792

Fang, Z., Wang, Y., Peng, L., & Hong, H. (2021). Predicting flood susceptibility using LSTM neural networks. *Journal of Hydrology*, *594*. https://doi.org/10.1016/j.jhydrol.2020.125734

FEMA. (1997). Multi Hazard Identification and Asessment. *Washington, DC, USA*.

Mtega, W. P., Bernard, R., Msungu, A. C., & Sanare, R. (2012). *Using mobile phones for teaching and learning purposes in higher learning institutions: The case of Sokoine University of Agriculture in Tanzania*.

Pedro, F., Subosa, M., Rivas, A., & Valverde, P. (2019). *Artificial intelligence in education: Challenges and opportunities for sustainable development*.

Pillay, N. (2020). Artificial intelligence for Africa: An opportunity for growth, development, and democratization. *University of Pretoria, Viewed*.

Rutenberg, I., Gwagwa, A., & Omino, M. (2021). Use and Impact of Artificial Intelligence on Climate Change Adaptation in Africa. In N. Oguge, D. Ayal, L. Adeleke, & I. da Silva (Eds.), *African Handbook of Climate Change Adaptation* (pp. 1107–1126). Springer International Publishing. https://doi.org/10.1007/978-3-030-45106-6_80

Schoeman, W., Moore, R., Seedat, Y., & Chen, J. Y.-J. (2021). *Artificial intelligence: Is South Africa ready?*

Shahabi, H., Shirzadi, A., Ronoud, S., Asadi, S., Pham, B. T., Mansouripour, F., Geertsema, M., Clague, J. J., & Bui, D. T. (2021). Flash flood susceptibility mapping using a novel deep learning model based on deep belief network, back propagation and genetic algorithm. *Geoscience Frontiers*, *12*(3). https://doi.org/10.1016/j.gsf.2020.10.007

UNISDR. (2023). *Disaster classification - Global Disaster Loss Collection Initiative*. https://desinventar.cimafoundation.org/disasterclassification.html

UNISDR, U. N. I. S. for D. (2009). UNISDR terminology on disaster risk reduction. *Geneva: United Nations*.

Vinuesa, R., Azizpour, H., Leite, I., Balaam, M., Dignum, V., Domisch, S., Felländer, A., Langhans, S. D., Tegmark, M., & Fuso Nerini, F. (2020). The role of artificial intelligence in achieving the Sustainable Development Goals. In *Nature Communications* (Vol. 11, Issue 1). Nature Research. https://doi.org/10.1038/s41467-019-14108-y

Wang, Y., Fang, Z., Hong, H., & Peng, L. (2020). Flood susceptibility mapping using convolutional neural network frameworks. *Journal of Hydrology*, *582*. https://doi.org/10.1016/j.jhydrol.2019.124482